# Pharmacogenetic allele variant frequencies: An analysis of the VA's Million Veteran Program (MVP) as a representation of the diversity in US population

**Kyriacos Markianos**[1,☉], **Frederic Dong**[1,☉], **Bryan Gorman**[1], **Yunling Shi**[1],
**Daniel Dochtermann**[1], **Uma Saxena**[1], **Poornima Devineni**[1], **Jennifer Moser**[2],
**Sumitra Muralidhar**[2], **Rachel Ramoni**[2], **Philip Tsao**[3], **Saiju Pyarajan**[1]*,
**Ronald Przygodzki**[2]*, **for the Million Veteran Program**[¶]

1 Center for Data and Computational Sciences, VA Boston HealthCare System, Boston, MA, United States of America, 2 The office of Research and Development, US Department of Veterans Affairs, Washington, DC, United States of America, 3 VA Palo Alto Health Care System, Palo Alto, CA, United States of America

☉ These authors contributed equally to this work.
¶ Complete membership of the author group can be found in the Acknowledgments.
* Ronald.Przygodzki@va.gov (RP); Saiju.Pyarajan@va.gov (SP)

**Data Availability Statement:** All relevant data are within the paper and its Supporting Information files.

## Abstract

We present allele frequencies of pharmacogenomics relevant variants across multiple ancestry in a sample representative of the US population. We analyzed 658,582 individuals with genotype data and extracted pharmacogenomics relevant single nucleotide variant (SNV) alleles, human leukocyte antigens (HLA) 4-digit alleles and an important copy number variant (CNV), the full deletion/duplication of *CYP2D6*. We compiled distinct allele frequency tables for European, African American, Hispanic, and Asian ancestry individuals. In addition, we compiled allele frequencies based on local ancestry reconstruction in the African-American (2-way deconvolution) and Hispanic (3-way deconvolution) cohorts.

## Introduction

Genetic polymorphisms of metabolic pathways and cytochrome P450 (CYP) genes are associated with altering pharmacokinetics and pharmacodynamics of the absorption, distribution, metabolism and excretion (ADME) of drug and toxic compounds (xenobiotics). Gaining a better understanding of the interindividual variations of this genetic makeup is necessary to understand the metabolic rate of efficiency a xenobiotic is metabolized. In general, heritable selective pressure is a major determinant of variant frequency among the different ethnic populations, typically presenting with two or more variants identified in most metabolic pathway genes. Common star allelic variants (referred to herein as "variant") prescribe a "normal" metabolic cycle while others convey a heightened or depressed metabolic cycle. Much of this is well catalogued in a variety of collections, including PharmGKB (**https://www.pharmgkb.org/**), with clinically actionable variant-vs-drug combinations presented in the Clinical Pharmacogenetics Implementation Consortium (**CPIC https://cpicpgx.org/**) and the Dutch

**Funding:** This work was supported by grant #MVP000 to SP from the Million Veteran Program (MVP) from the Veterans Affairs (VA) Office of Research and Development (ORD) (www.research. va.gov). The funders had no role in study design, data collection and analysis, decision to publish, or preparation of the manuscript.

**Competing interests:** The authors have declared that no competing interests exist.

Pharmacogenetics Working Group (**DPWG http://upgx.eu/guidelines**). While these variants are catalogued in the multitude of databases, it is also important to recognize that many of the variants identified heavily rely upon data derived from unique ethnic populations. Ethnic population data are typically derived from a limited collection of self-identified subjects and the unique variants associated within that ethnicity. Moreover, certain variants designated as normal are unique to a select ethnicity and not represented among others, such as is known for *CYP2D6* and codeine, or *CYP3A5* and Tacrolimus [1]. Lastly, while certain populations are considered relatively homogenous over several generations as dictum of culture, not all data is reflective of this consideration which further contributes to the diversity of drug responses.

The distribution of inherited xenobiotic-metabolizing alleles differs considerably between populations [2, 3] and appears to be rigid in frequency among ethnically stable populations. While there are several large-scale data sets that can provide variant frequencies of pharmacogenomic genes for researchers and clinicians to use, most of these data are not representative of the "melting pot" of the genetic ancestries present within the United States. (US). While one could rely on self-reported ethnicity to improve variant frequency found among unique populations, such data is imperfect [4]. Further complicating possible variant predictions is that nearly everyone has at least one pharmacogenomic variant allele with as many as 3% carrying 5 allelic variants [5]. These findings limit the overarching use of ethnically related variant frequencies in diverse populations such as is present in the US. This is because the data available is limited to a specific self-reported ethnicity and/or does not consider other variants that could be present among other ethnicities. This is a particularly important consideration for research of personalized drug therapy and potentially changes the healthcare guidelines provided by groups such as CPIC and DPWG that select important alleles for clinical genotyping based in part on population prevalence.

To address the issues with the use of pharmacogenomic variants and to further explore possibly pharmacogenetically-associated variant markers we used the Million Veteran Program (MVP) [6] with >800,000 participants to generate a coherent representation of allelic frequencies present within a US population. The MVP cohort is mostly male but is very diverse and represents the US population ancestry in general. The genotype data was imputed using the African Genome Resources (AGR) and 1000 Genomes imputation reference panels.

## Results

Our analysis is based on the Release 4 of MVP data with 658,582 individuals genotyped with the MVP-1 Axiom array [7]. Participants were assigned ancestry based on the HARE algorithm (Harmonized Ancestry and Race/Ethnicity) [8]. The MVP cohort is diverse with ~30% of the cohort assigned as non-European (EUR 467k, AFR 125k, HIS 52k, ASN 8k). A small fraction of the cohort was highly admixed and not assigned to any of the four major ancestries and is not included in this analysis (<2%).

Our aim is to provide ancestry specific variant frequency catalog for a significant fraction of pharmacogenomics relevant variants in a large cohort representative of the US population. We examined Single Nucleotide Variants (SNV), an important pharmacogenetics relevant Copy Number Variant (CNV) as well as Human Leukocyte Antigen (HLA) 4-digit alleles. We defined our pharmacogenomics gene set by combining information from two publicly available data bases, PharmGKB and PharmVar (Methods). Overall, we were able to determine SNV frequencies for 273/1339 targeted SNVs, in 148/152 targeted genes. Details on variant selection can be found in Methods. S1 File provides a comprehensive table of all allele frequencies. As expected, SNV allele frequencies vary substantially among HARE groups (Fig 1).

a

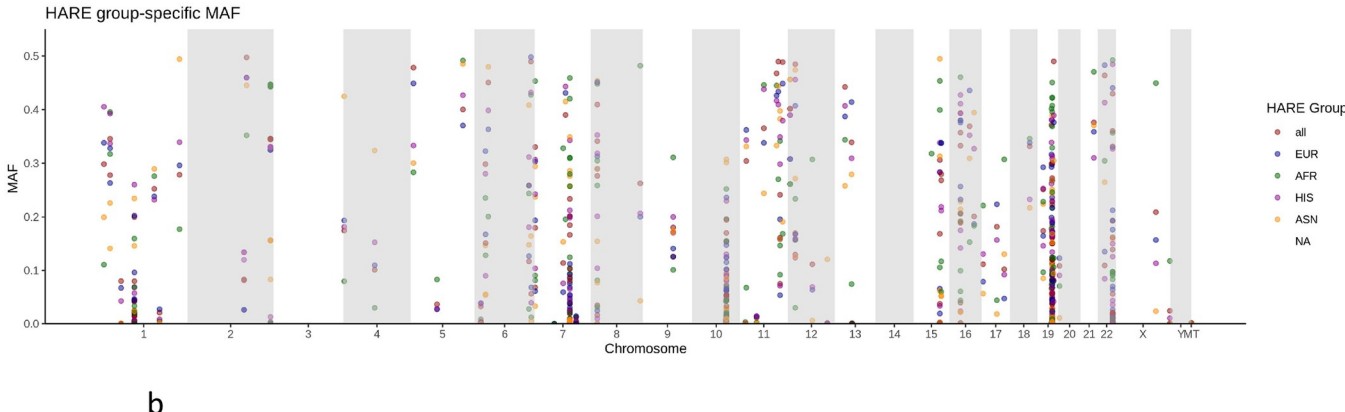

b

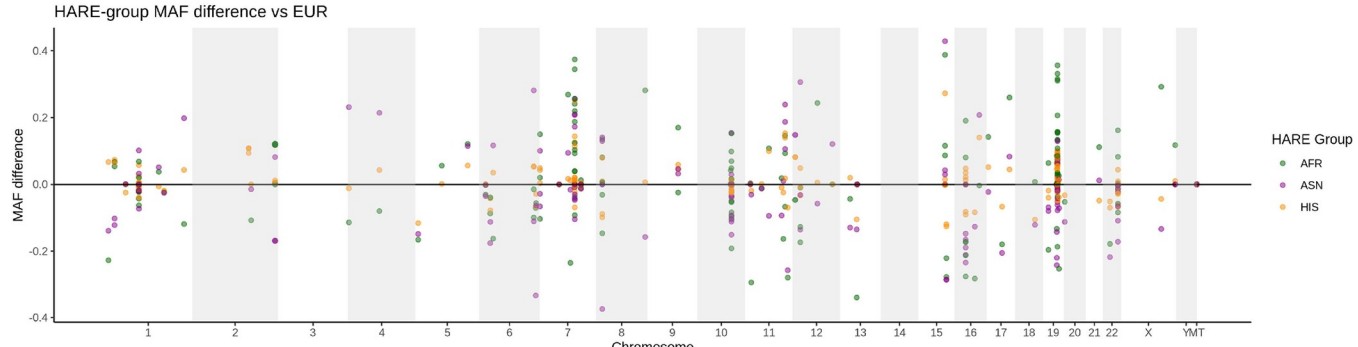

**Fig 1.** (a) Allele frequency distributions in different ethnic groups of 282 variants in 153 genes (b) differences in allele frequencies relative to EUR samples.

In addition to HARE allele frequencies, we used Local Ancestry Inference (LAI) to identify ancestral origin of individual chromosomal segments and compute allele frequencies based on the local ancestry. We "painted" the African American samples (125 k individuals) using two-way deconvolution, extracting allele frequencies for the AFR and EUR tracks. For the Hispanic individuals (52 k) we used three-way deconvolution to compute allele frequencies for EUR, AFR and Native American (AMR) tracks. Details on the LAI will be presented elsewhere. In Fig 2 we present allele frequencies derived from HARE groups, LAI as well as two publicly available databases, 1 k genome and gnomAD (Methods). The most striking differences are observed for Hispanics, a group that is extremely heterogeneous and not well defined in the genetics literature.

Allele frequencies for the three major MVP HARE groups (EUR, AFR, HIS) are in good agreement with gnomAD derived estimates. However, comparison of gnomAD HIS frequencies with the LAI AMR track of the MVP HIS population reveals significant differences (S1 Fig). Here we note that the LAI AMR track provides much better allele frequency

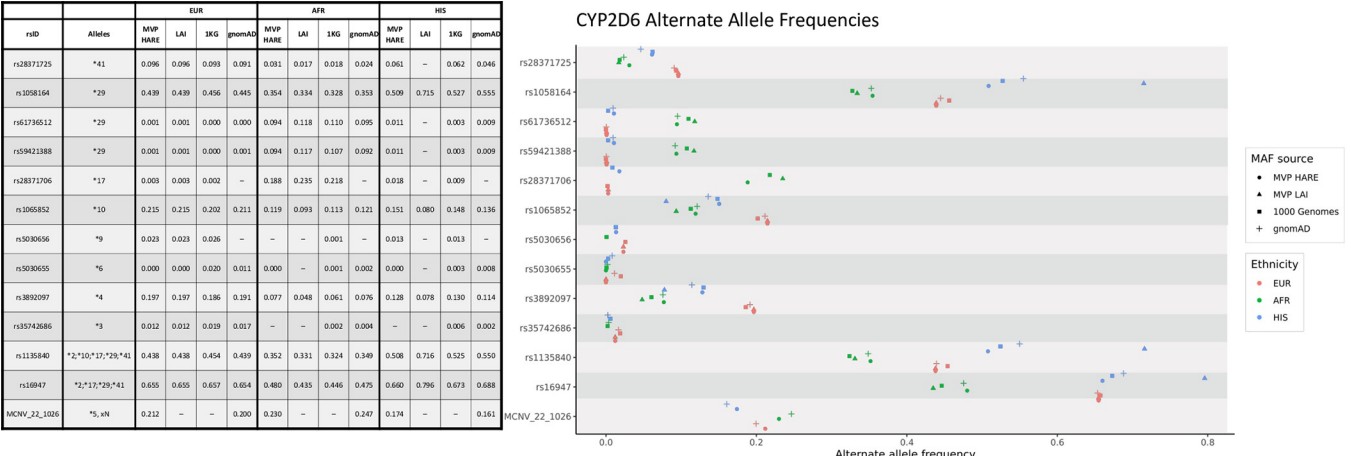

| rsID | Alleles | EUR | | | | AFR | | | | HIS | | | |
|---|---|---|---|---|---|---|---|---|---|---|---|---|---|
| | | MVP HARE | LAI | 1KG | gnomAD | MVP HARE | LAI | 1KG | gnomAD | MVP HARE | LAI | 1KG | gnomAD |
| rs28371725 | *41 | 0.096 | 0.096 | 0.093 | 0.091 | 0.031 | 0.017 | 0.018 | 0.024 | 0.061 | – | 0.062 | 0.046 |
| rs1058164 | *29 | 0.439 | 0.439 | 0.456 | 0.445 | 0.354 | 0.334 | 0.328 | 0.353 | 0.509 | 0.715 | 0.527 | 0.555 |
| rs61736512 | *29 | 0.001 | 0.001 | 0.000 | 0.000 | 0.094 | 0.118 | 0.110 | 0.095 | 0.011 | – | 0.003 | 0.009 |
| rs59421388 | *29 | 0.001 | 0.001 | 0.000 | 0.001 | 0.094 | 0.117 | 0.107 | 0.092 | 0.011 | – | 0.003 | 0.009 |
| rs28371706 | *17 | 0.003 | 0.003 | 0.002 | – | 0.188 | 0.235 | 0.218 | – | 0.018 | – | 0.009 | – |
| rs1065852 | *10 | 0.215 | 0.215 | 0.202 | 0.211 | 0.119 | 0.093 | 0.113 | 0.121 | 0.151 | 0.080 | 0.148 | 0.136 |
| rs5030656 | *9 | 0.023 | 0.023 | 0.026 | – | – | – | 0.001 | – | 0.013 | – | 0.013 | – |
| rs5030655 | *6 | 0.000 | 0.000 | 0.020 | 0.011 | 0.000 | – | 0.001 | 0.002 | 0.000 | – | 0.003 | 0.008 |
| rs3892097 | *4 | 0.197 | 0.197 | 0.186 | 0.191 | 0.077 | 0.048 | 0.061 | 0.076 | 0.128 | 0.078 | 0.130 | 0.114 |
| rs35742686 | *3 | 0.012 | 0.012 | 0.019 | 0.017 | – | – | 0.002 | 0.004 | – | – | 0.006 | 0.002 |
| rs1135840 | *2;*10;*17;*29;*41 | 0.438 | 0.438 | 0.454 | 0.439 | 0.352 | 0.331 | 0.324 | 0.349 | 0.508 | 0.716 | 0.525 | 0.550 |
| rs16947 | *2;*17;*29;*41 | 0.655 | 0.655 | 0.657 | 0.654 | 0.480 | 0.435 | 0.446 | 0.475 | 0.660 | 0.796 | 0.673 | 0.688 |
| MCNV_22_1026 | *5, xN | 0.212 | – | – | 0.200 | 0.230 | – | – | 0.247 | 0.174 | – | – | 0.161 |

**Fig 2. CYP2D6 allele frequencies for three MVP HARE groups, MVP Local Ancestry Inference (LAI) and two publicly available data sets: 1000 genome project and gnomAD.** Only consensus Tier 1 alleles [9] are shown. We note that we did not perform LAI in HARE EUR samples. Thus LAI frequencies are identical by default to HARE EUR frequencies. For HARE HIS, LAI corresponds to the AMR track allele frequencies (3-way deconvolution). We estimate imputation quality per site and ethnicity. We do not present allele frequencies for poorly imputed sites.

ascertainment than the 60 AMR genomes that were used to anchor the local ancestry deconvolution. In the MVP HARE HIS population (52k individuals) the AMR track contributes ~30% of the genome resulting in an effective population size of ~15k individuals. Furthermore, while the 60 AMR genomes we used to anchor ancestry deconvolution provide sufficient multi-locus information to resolve local ancestry, the AMR track is a much better sampling of the AMR genome as it exists today in the US population.

Sirolimus is a widely used immunosuppressant and the variant controlling its metabolism, rs2242480 (allele *CYP3A4*1G*) [10], varies among populations. We observe widely different allele frequencies in the three major groups (EUR, AFR, HIS) for gnomAD (0.09, 0.74, 0.37) and MVP (0.09, 0.73, 0.35). However, the local ancestry derived AMR allele frequency (0.67) is almost twice as high as the HIS allele frequency. The same observation applies to tacrolimus, another significant immunosuppressant. The controlling variant, rs776746 (*CYP3A5*3*), shows large variation in major MVP groups (0.07, 0.70, 0.21) and there is a significant difference between HIS and local ancestry derived AMR allele frequency (0.31). Thus, recent demographic history of individuals, and the fraction of inheritance derived from different major population groups, has a large impact on the allele frequency distribution of pharmacogenomics relevant variants.

*CYP2D6* is an important component of cytochrome P450 and is involved in the metabolism of many commonly prescribed medications, including antidepressants, antipsychotics, beta-blockers, opioids, antiemetics, atomoxetine, and tamoxifen [9, 11]. In addition to SNV frequencies for the most significant variants [12], Fig 2 presents allele frequencies for an important copy number variant, the whole gene deletion designated as *CYP2D6*5* in the pharmacogenomics literature (Figs 2 and 3). We called the *CYP2D6*5* CNV using UMAP, a machine learning algorithm [13]. Assignments are clearly separated for copy gain and copy loss. Furthermore, we can clearly separate single and double copy loss (Fig 3). The UMAP approach offers a clear advantage over classification based on Principal Components Analysis (PCA, S2 Fig). We note that UMAP does not represent a general approach to copy number variation detection. Hyper-parameters for the model are tuned for the specific, relatively common CNVs. Furthermore, we achieve optimal performance only when we tune the model separately for individual HARE ancestries.

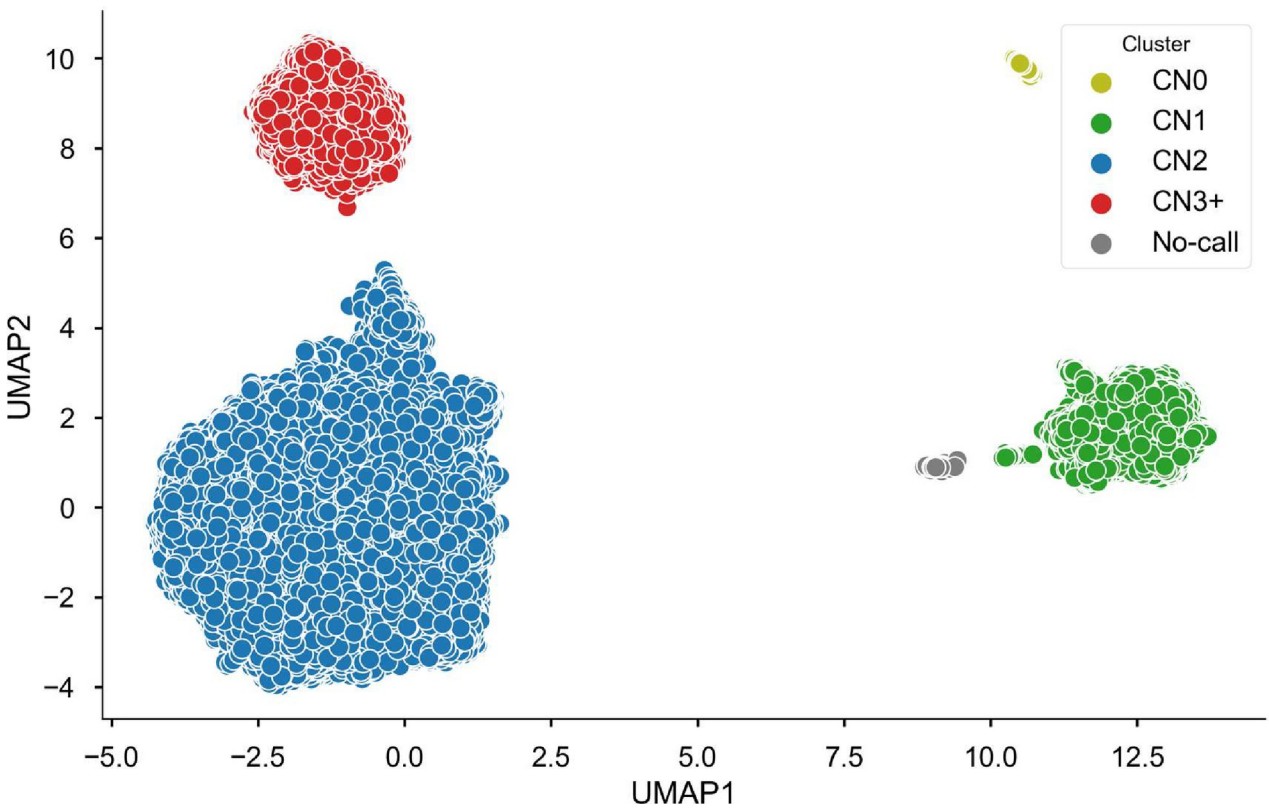

**Fig 3. Copy number variation in CYP2D6.** Results are shown just for the HARE AFR cohort; clusters were derived using UMAP [13].

Table 1A presents *CYP2D6*\*5* allele frequencies for the three major HARE groups. The major survey of *CYP2D6*\*5* [14] finds slightly different allele frequencies, e.g., 89% in Beoris et al vs 78% in MVP for copy number 2 EUR. Our findings are closer to the frequencies reported by gnomAD (80%, MCNV_22_1026 | gnomAD SVs v2.1 | gnomAD (broadinstitute.org). The differences might be due to different assays: single site PCR vs SNP genotyping (MVP) or sequencing (gnomAD), or differences in ascertainment of ethnic background. We are not able to run the UMAP algorithm on phased chromosomes, but we can use ancestry deconvolution and test CNV status in individuals ancestry-homozygous at CYP2D6, e.g. AMR/AMR individuals in the HARE HIS group. Results are shown in Table 1B. The ancestral AMR genome harbors fewer single-copy samples while the EUR tracks are in close agreement with observations in the EUR HARE cohort.

In addition to SNVs and specific CNVs we derived HLA alleles from SNP genotypes using the HIBAG algorithm [15] (HLA Imputation using attribute BAGging). Although HLA status

**Table 1. Allele frequencies for allele *CYP2D6*\*5* (whole gene deletion) in different HARE groups.** In addition, we show allele frequencies in the three components of the HARE HIS group (EUR, AFR, AMR) calculated using individuals ancestry-homozygous at *CYP2D6*.

| Copies | HARE group | | | HARE HIS, ancestry homozygous segments | | |
|---|---|---|---|---|---|---|
| | EUR | AFR | HIS | EUR | AFR | AMR |
| 0 | 0.14 | 0.37 | 0.14 | 0.16 | 0.31 | 0.03 |
| 1 | 6.25 | 10.70 | 5.71 | 5.93 | 9.87 | 4.47 |
| 2 | 78.40 | 76.90 | 82.50 | 77.01 | 79.78 | 93.05 |
| 3+ | 14.80 | 11.96 | 11.58 | 16.89 | 10.03 | 2.45 |

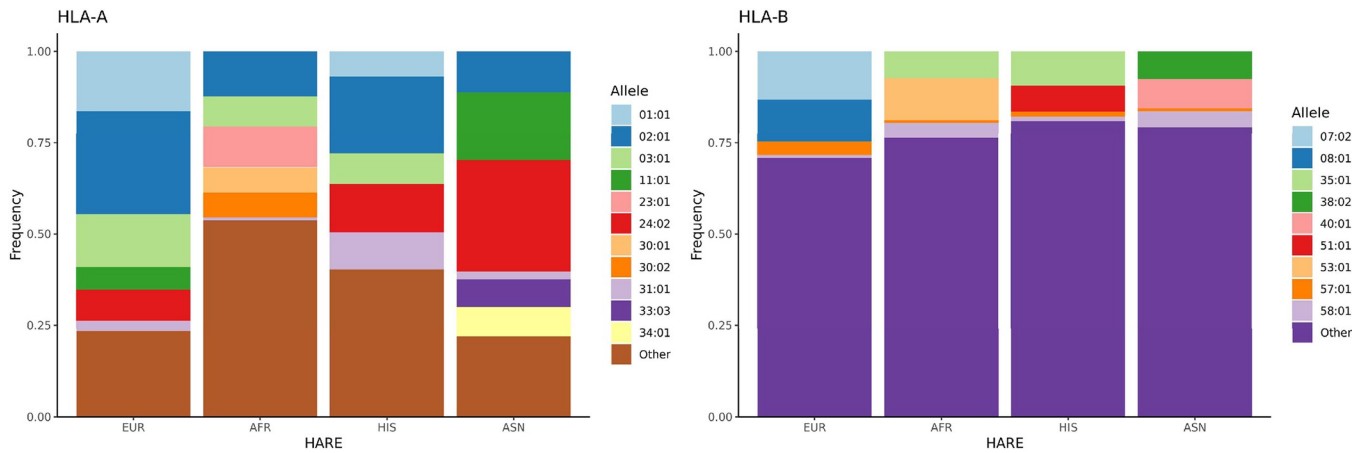

**Fig 4. HLA allele distribution in 4 ethnic groups.** Allele imputation was performed through HIBAG using the Axiom UK Biobank model and Axiom MVP genotypes.

does not modify pharmacokinetics there are well established adverse drug reactions in the presence of specific HLA alleles. For example, abacavir, a common anti-retroviral, causes abacavir hypersensitivity syndrome in the presence of *HLA-B*5701*; Allopurinol is typically a safe drug for the treatment of gout but in the presence of *HLA-B*5801* is associated with an increased risk for allopurinol induced severe cutaneous adverse drug reaction (SCAR) with most serious cases developing Stevens–Johnson syndrome and toxic epidermal necrolysis (SJS/TEN) [16]. HLA 4-digit Class I and Class II allele distribution for four HARE groups is shown in Fig 4. As expected, allele frequencies are highly variable in the four groups, including the three alleles most relevant for pharmacogenomics: *HLA-A*3101*, *HLA-B*5701* and *HLA-B*58:01* (Table 2). Details of HLA allele imputation will be presented elsewhere. Here, we note that HLA imputation precision was >90% for HARE EUR, AFR and HIS groups. However, we currently observe lower precision for the ASN predictions due to lack of an appropriate training set.

## Discussion

We present a survey of pharmacogenetics relevant variants in the MVP, a sample representative of the US population. Using the MVP-1 Axiom array we can resolve a large fraction of known pharmacogenomics alleles, either as direct or as imputed genotypes. In addition, we use the genotypes to derive population allele frequencies for an important common CNV, whole gene deletion/duplication of *CYP2D6*, as well as population distribution of HLA alleles, including HLA alleles important for drug delivery decisions.

As expected, there is substantial variation in allele frequencies between ancestry groups for a subset of the examined variants. In addition to allele frequencies of individual ancestry groups (HARE EUR, AFR, HIS, ASN) we use an innovative approach, LAI, to derive allele

**Table 2. Allele frequencies for HLA alleles with known large effects in pharmacogenomics.**

| Allele | EUR | AFR | HIS | ASN |
|---|---|---|---|---|
| HLA-A*31:01 | 2.75 | 0.82 | 5.08 | 2.21 |
| HLA-B*57:01 | 3.73 | 0.79 | 1.41 | 0.75 |
| HLA-B*58:01 | 0.68 | 4.07 | 1.23 | 4.45 |

frequencies of ancestral genomes present in recently admixed US populations; 2-way deconvolution for HARE AFR (EUR, AFR) and 3-way deconvolution for HARE HIS (EUR, AFR, AMR).

LAI conveys important information on allele frequency distribution in under-represented populations. Allele frequency is an important consideration in the formulation of clinical genotyping guidelines provided by groups such as CPIC and DPWG. The AMR track is a much better sampling of the AMR genome as it exists today in the US population compared to the small, and not necessarily representative, number of samples from AMR populations (60 vs 15,000 effective genomes). Sites with significantly different allele frequencies in AMR and EUR/AFR tracks are sites where self-identification as HIS provides limited power to guess likelihood of drug sensitivity. Thus, they are sites where groups such as CPIC and DPWG should rely on LAI minimum allele frequency rather than ethnic group allele frequency for recommendations. The large sample size we use for our analysis is particularly important for low frequency variants. For example, single and double-copy deletions of CYP2D6 are relatively rare. Therefore, it is inappropriate to derive frequencies from a reference a panel, even under the assumption that the reference panel is representative of the general US population.

There are limitations in our derived population allele frequencies. While SNVs are phased neither CNV nor HLA calls are phased genotypes. Furthermore, successful phasing in the overall genome does not guarantee successful phasing in complex genomic regions such as *CYP2D6*. For *CYP2D6* in particular, we have been able to resolve whole gene deletions and duplications, but we are certain that there is additional small scale copy number variation that cannot be resolved by our UMAP machine learning approach. For example, small deletions and complex rearrangements involving the proximal *CYP2D7* and *CYP2D8* pseudogenes. It is likely that such complex variation has a minor contribution to the population distribution of *CYP2D6* pharmacogenetics. However, resolution of population level frequency of such variants will require specialized assays such as long-range sequencing. Improving phasing and imputation will aid the eventual derivation of star alleles in these regions.

We think that this comprehensive allele frequency report in a population representative of the US genome diversity will become a useful reference for future guidelines of relative importance of alleles worth ascertaining in pharmacogenetics screens. High variance of allele frequency, not only among ethnic groups but most importantly among ancestral genomes contributing to mixed ancestry individuals in the US population, further underscores the need for individual typing rather than reliance on self-reported ethnicity on drug delivery decisions in clinical practice. We hope this manuscript promotes the adoption of personalized medicine in under-represented populations.

## Methods

### Ethics statement

The Veterans Affairs (VA) central institutional review board (cIRB) and site-specific IRBs approved the Million Veteran Program study.

### MVP genotype data

The MVP Release 4 dataset includes 658,582 individuals and consists of a hard-called dataset of 667,955 variants prepared as described in Hunter-Zinck et al. 2020 [7], as well as an imputed dataset. Genotype calls passing initial quality control were further prepared for phasing and imputation by removing markers with high missingness (>20%), monomorphic markers, and markers significantly out of Hardy-Weinberg equilibrium (p < 1e-6 adjusted for ancestry). Haplotypes were then statistically phased using SHAPEIT v4.1.3 (https://odelaneau.github.io/shapeit4/) and

imputed into the African Genome Resources and 1000 Genomes imputation panels using Minimac4 (https://genome.sph.umich.edu/wiki/Minimac4). Each individual in the cohort was assigned a HARE group (EUR, AFR, HIS, or ASN), a surrogate variable for ancestry and race/ethnicity (Fang et al. 2019). The MVP Release 4 cohort consists of 467,162 EUR, 124,756 AFR, 52,423 HIS, 8,364 ASN, and 5,877 unassigned individuals. All analysis was performed in GRCh37.

## Identification of known pharmacogenetics variants

We curated a catalogue of known or high-confidence pharmacogenetics variants by rsID from the PharmGKB and PharmVar databases. From PharmGKB, we downloaded variant summary data (https://api.pharmgkb.org/v1/download/file/data/variants.zip) and kept only variants with at least one Level 1 or 2 PharmGKB clinical annotation. From PharmVar, we downloaded the complete database (version 4.2.4) and kept all variants. In total, we identify 1,339 unique variants from 152 genes.

## Identification of pharmacogenetics variants in the MVP genotype dataset

**Genotyped dataset.**   We selected the intersection of known pharmacogenetics variants with the catalog of SNPs in the MVP array. We identified pharmacogenetics variants by chromosome location and rsID [7].

**Imputed dataset.**   Imputation was performed using MINIMAC. We kept only variants with imputation $R^2 > 0.9$ within the ethnic group. We assigned rsIDs to imputed variants by intersecting variant genomic position with rsID genomic position in NCBI dbSNP (v154) using bedtools. We then identified pharmacogenetics variants by overlapping imputed variant rsIDs.

In total, we find 193 pharmacogenetics variants from 136 genes in the genotyped data set. Including the imputed variants, we expand the set to 273 variants in 148 genes. If we relax the selection criteria to include all imputed variants that satisfy imputation $R^2 > 0.9$ in any one of the 4 HARE groups (EUR, AFR, HIS, ASN) we expand the set to 408 variants.

## Allele frequency analyses

**Calculation of minor allele frequencies.**   HARE group-specific minor allele frequencies (MAFs) were calculated for the MVP hard-called and imputed datasets using PLINK2.

**Local Ancestry Inference (LAI) based allele frequencies.**   Briefly, we performed LAI using rfmix2. We used 3,942 reference samples for EUR, AFR and Native American (AMR) ancestry collected by the 1000 genome project and the Human Genome Diversity Project (HGDP). The reference VCF files were curated by the gnomAD team (https://gnomad.broadinstitute.org/downloads#v3-hgdp-1kg). We used local ancestry output to create separate, ancestry specific, VCF output files. Two files for the HARE AFR sample (EUR-AFR) and three files for the HARE Hispanic sample (EUR-AFR-NAT). The allele frequency extraction procedure was the same for LAI and gnomAD samples, described below.

**GnomAD allele frequencies.**   Population-specific frequencies were extracted as follows. LAI and gnomAD (v2.1.1 Genomes only, not Exomes) frequencies were stored in the INFO fields of VCF files. AFR and HIS LAI frequencies were stored in separate files. gnomAD frequencies were stored in population-specific INFO fields (AF_nfe, AF_afr, AF_amr for non-Finnish Europeans, African/African Americans, and Latino/Admixed Americans respectively). Using bcftools 1.10, VCF files were first filtered to the relevant SNPs (bcftools view—include 'ID = @<file of rsIDs> ' <VCF file>), and frequencies were then extracted from the relevant INFO fields (e.g., bcftools query -'%ID\t%INFO/AF_nfe\n').

**1000 genomes allele frequencies.** 1000 Genomes population-specific MAFs were extracted from 1000 Genomes Phase 3 VCFs.

## Analysis and visualization

Visualization of MAFs, and calculation and visualization of MAF differences between MVP and 1000 Genomes, was performed using R.

## HLA type predictions

4-digit HLA type predictions were generated for HLA-A and HLA-B from hard-called genotype data using HIBAG [15]. We chose the pre-fit Affymetrix Axiom UK Biobank Array 4-digit resolution model (https://hibag.s3.amazonaws.com/hlares_index.html), as the MVP genotyping array covers $> 95\%$ of this model's training variants for both loci. We used the European model for individuals assigned to HARE group EUR and the multi-ethnic model for individuals assigned to HARE groups AFR, HIS, and ASN. Predictions were generated by calling the *predict()* function from HIBAG. Frequencies were calculated for each 4-digit allele by HARE group.

## Supporting information

**S1 Fig. Allele frequency comparisons between gnomAD and MVP HARE groups for three groups (EUR, AFR, HIS).** For all three, correlation with gnomAD allele frequencies is high ($R^2>0.99$). In the lower right we compare allele frequencies for gnomAD HIS and Local Ancestry Inference (LAI) derived allele frequencies for the AMR track of the HARE HIS group ($R^2 = 0.91$). We use three-way local ancestry deconvolution (EUR, AFR, AMR).
(TIF)

**S2 Fig. Copy number variation in CYP2D6 using two computational approaches.** Results are shown just for the HARE AFR cohort; clusters were derived using (a) Principal Components Analysis (PCA) and (b) UMAP(13). UMAP significantly reduces assignment ambiguity.
(TIF)

**S1 File. Allele frequency table for HARE groups and LAI tracks.** In addition to allele frequencies, we provide imputation quality information per site and ethnicity/LAI-track (imputation R2). In the same table we attach PharmGKB annotation per site, where available. In the table we include all sites with imputation $R^2 > .9$ in ANY of the four HARE groups (EUR, AFR, HIS, ASN) for a total of 408 sites. Four sites are multiallelic, thus the table has 412 rows.
(CSV)

## Acknowledgments

This research is based on data from the Million Veteran Program, Office of Research and Development, Veterans Health Administration. This publication does not represent the views of the Department of Veterans Affairs, the US Food and Drug Administration, or the US Government.

   Million Veteran Program Full Acknowledgments
   MVP Executive Leadership

- * Sumitra Muralidhar, Ph.D. (Director, MVP)

   US Department of Veterans Affairs, 810 Vermont Avenue NW, Washington, DC 20420

- J. Michael Gaziano, M.D., M.P.H.

VA Boston Healthcare System, 150 S. Huntington Avenue, Boston, MA 02130

- Philip S. Tsao, Ph.D.

  VA Palo Alto Health Care System, 3801 Miranda Avenue, Palo Alto, CA 94304
  MVP Program Office

- Program Director—Sumitra Muralidhar, Ph.D.

  US Department of Veterans Affairs, 810 Vermont Avenue NW, Washington, DC 20420

- Associate Director, Scientific Programs

- Jennifer Moser, Ph.D.

  US Department of Veterans Affairs, 810 Vermont Avenue NW, Washington, DC 20420

- Associate Director, Cohort Management & Public Relations

- Jennifer E. Deen, B.S.

  US Department of Veterans Affairs, 810 Vermont Avenue NW, Washington, DC 20420
  MVP Operations

- Director of Regulatory Affairs–Lori Churby, B.S.

  VA Palo Alto Health Care System, 3801 Miranda Avenue, Palo Alto, CA 94304

- MVP Cohort Management Director–Stacey B. Whitbourne, Ph.D.

  VA Boston Healthcare System, 150 S. Huntington Avenue, Boston, MA 02130

- MVP Recruitment/Enrollment Director—Jessica V. Brewer, M.P.H.

  VA Boston Healthcare System, 150 S. Huntington Avenue, Boston, MA 02130

- Director, VA Central Biorepository, Boston–Mary T. Brophy M.D., M.P.H.

  VA Boston Healthcare System, 150 S. Huntington Avenue, Boston, MA 02130

- MVP Informatics, Boston–Shahpoor (Alex) Shayan, M.S.

  VA Boston Healthcare System, 150 S. Huntington Avenue, Boston, MA 02130

- Director, Center for Computational and Data Sciences (C-DACS) & Genomics Core–Saiju Pyarajan Ph.D.

  VA Boston Healthcare System, 150 S. Huntington Avenue, Boston, MA 02130

- Director, Phenomics Data Core–Kelly Cho, M.P.H, Ph.D.

  VA Boston Healthcare System, 150 S. Huntington Avenue, Boston, MA 02130
  Current MVP Local Site Investigators

- Atlanta VA Medical Center (Peter Wilson, M.D.)

  1670 Clairmont Road, Decatur, GA 30033

- Bay Pines VA Healthcare System (Rachel McArdle, Ph.D.)

  10,000 Bay Pines Blvd Bay Pines, FL 33744

- Birmingham VA Medical Center (Louis Dellitalia, M.D.)

  700 S. 19th Street, Birmingham AL 35233

- Central Western Massachusetts Healthcare System (Kristin Mattocks, Ph.D., M.P.H.)

  421 North Main Street, Leeds, MA 01053

- Cincinnati VA Medical Center (John Harley, M.D., Ph.D.)

  3200 Vine Street, Cincinnati, OH 45220

- Clement J. Zablocki VA Medical Center (Jeffrey Whittle, M.D., M.P.H.)

  5000 West National Avenue, Milwaukee, WI 53295

- VA Northeast Ohio Healthcare System (Frank Jacono, M.D.)

  10701 East Boulevard, Cleveland, OH 44106

- Durham VA Medical Center (Jean Beckham, Ph.D.)

  508 Fulton Street, Durham, NC 27705

- Edith Nourse Rogers Memorial Veterans Hospital (John Wells., Ph.D.)

  200 Springs Road, Bedford, MA 01730

- Edward Hines, Jr. VA Medical Center (Salvador Gutierrez, M.D.)

  5000 South 5th Avenue, Hines, IL 60141

- Veterans Health Care System of the Ozarks (Kathrina Alexander, M.D.)

  1100 North College Avenue, Fayetteville, AR 72703

- Fargo VA Health Care System (Kimberly Hammer, Ph.D.)

  2101 N. Elm, Fargo, ND 58102

- VA Health Care Upstate New York (James Norton, Ph.D.)

  113 Holland Avenue, Albany, NY 12208

- New Mexico VA Health Care System (Gerardo Villareal, M.D.)

  1501 San Pedro Drive, S.E. Albuquerque, NM 87108

- VA Boston Healthcare System (Scott Kinlay, M.B.B.S., Ph.D.)

  150 S. Huntington Avenue, Boston, MA 02130

- VA Western New York Healthcare System (Junzhe Xu, M.D.)

  3495 Bailey Avenue, Buffalo, NY 14215–1199

- Ralph H. Johnson VA Medical Center (Mark Hamner, M.D.)

  109 Bee Street, Mental Health Research, Charleston, SC 29401

- Columbia VA Health Care System (Roy Mathew, M.D.)

  6439 Garners Ferry Road, Columbia, SC 29209

- VA North Texas Health Care System (Sujata Bhushan, M.D.)

  4500 S. Lancaster Road, Dallas, TX 75216

- Hampton VA Medical Center (Pran Iruvanti, D.O., Ph.D.)

100 Emancipation Drive, Hampton, VA 23667

- Richmond VA Medical Center (Michael Godschalk, M.D.)

  1201 Broad Rock Blvd., Richmond, VA 23249

- Iowa City VA Health Care System (Zuhair Ballas, M.D.)

  601 Highway 6 West, Iowa City, IA 52246–2208

- Eastern Oklahoma VA Health Care System (River Smith, Ph.D.)

  1011 Honor Heights Drive, Muskogee, OK 74401

- James A. Haley Veterans' Hospital (Stephen Mastorides, M.D.)

  13000 Bruce B. Downs Blvd, Tampa, FL 33612

- James H. Quillen VA Medical Center (Jonathan Moorman, M.D., Ph.D.)

  Corner of Lamont & Veterans Way, Mountain Home, TN 37684

- John D. Dingell VA Medical Center (Saib Gappy, M.D.)

  4646 John R Street, Detroit, MI 48201

- Louisville VA Medical Center (Jon Klein, M.D., Ph.D.)

  800 Zorn Avenue, Louisville, KY 40206

- Manchester VA Medical Center (Nora Ratcliffe, M.D.)

  718 Smyth Road, Manchester, NH 03104

- Miami VA Health Care System (Ana Palacio, M.D., M.P.H.)

  1201 NW 16th Street, 11 GRC, Miami FL 33125

- Michael E. DeBakey VA Medical Center (Olaoluwa Okusaga, M.D.)

  2002 Holcombe Blvd, Houston, TX 77030

- Minneapolis VA Health Care System (Maureen Murdoch, M.D., M.P.H.)

  One Veterans Drive, Minneapolis, MN 55417

- N. FL/S. GA Veterans Health System (Peruvemba Sriram, M.D.)

  1601 SW Archer Road, Gainesville, FL 32608

- Northport VA Medical Center (Shing Shing Yeh, Ph.D., M.D.)

  79 Middleville Road, Northport, NY 11768

- Overton Brooks VA Medical Center (Neeraj Tandon, M.D.)

  510 East Stoner Ave, Shreveport, LA 71101

- Philadelphia VA Medical Center (Darshana Jhala, M.D.)

  3900 Woodland Avenue, Philadelphia, PA 19104

- Phoenix VA Health Care System (Samuel Aguayo, M.D.)

  650 E. Indian School Road, Phoenix, AZ 85012

- Portland VA Medical Center (David Cohen, M.D.)

  3710 SW U.S. Veterans Hospital Road, Portland, OR 97239

- Providence VA Medical Center (Satish Sharma, M.D.)

  830 Chalkstone Avenue, Providence, RI 02908

- Richard Roudebush VA Medical Center (Suthat Liangpunsakul, M.D., M.P.H.)

  1481 West 10th Street, Indianapolis, IN 46202

- Salem VA Medical Center (Kris Ann Oursler, M.D.)

  1970 Roanoke Blvd, Salem, VA 24153

- San Francisco VA Health Care System (Mary Whooley, M.D.)

  4150 Clement Street, San Francisco, CA 94121

- South Texas Veterans Health Care System (Sunil Ahuja, M.D.)

  7400 Merton Minter Boulevard, San Antonio, TX 78229

- Southeast Louisiana Veterans Health Care System (Joseph Constans, Ph.D.)

  2400 Canal Street, New Orleans, LA 70119

- Southern Arizona VA Health Care System (Paul Meyer, M.D., Ph.D.)

  3601 S 6th Avenue, Tucson, AZ 85723

- Sioux Falls VA Health Care System (Jennifer Greco, M.D.)

  2501 W 22nd Street, Sioux Falls, SD 57105

- St. Louis VA Health Care System (Michael Rauchman, M.D.)

  915 North Grand Blvd, St. Louis, MO 63106

- Syracuse VA Medical Center (Richard Servatius, Ph.D.)

  800 Irving Avenue, Syracuse, NY 13210

- VA Eastern Kansas Health Care System (Melinda Gaddy, Ph.D.)

  4101 S 4th Street Trafficway, Leavenworth, KS 66048

- VA Greater Los Angeles Health Care System (Agnes Wallbom, M.D., M.S.)

  11301 Wilshire Blvd, Los Angeles, CA 90073

- VA Long Beach Healthcare System (Timothy Morgan, M.D.)

  5901 East 7th Street Long Beach, CA 90822

- VA Maine Healthcare System (Todd Stapley, D.O.)

  1 VA Center, Augusta, ME 04330

- VA New York Harbor Healthcare System (Peter Liang, M.D., M.P.H.)

  423 East 23rd Street, New York, NY 10010

- VA Pacific Islands Health Care System (Daryl Fujii, Ph.D.)

459 Patterson Rd, Honolulu, HI 96819

- VA Palo Alto Health Care System (Philip Tsao, Ph.D.)

  3801 Miranda Avenue, Palo Alto, CA 94304–1290

- VA Pittsburgh Health Care System (Patrick Strollo, Jr., M.D.)

  University Drive, Pittsburgh, PA 15240

- VA Puget Sound Health Care System (Edward Boyko, M.D.)

  1660 S. Columbian Way, Seattle, WA 98108–1597

- VA Salt Lake City Health Care System (Jessica Walsh, M.D.)

  500 Foothill Drive, Salt Lake City, UT 84148

- VA San Diego Healthcare System (Samir Gupta, M.D., M.S.C.S.)

  3350 La Jolla Village Drive, San Diego, CA 92161

- VA Sierra Nevada Health Care System (Mostaqul Huq, Pharm.D., Ph.D.)

  975 Kirman Avenue, Reno, NV 89502

- VA Southern Nevada Healthcare System (Joseph Fayad, M.D.)

  6900 North Pecos Road, North Las Vegas, NV 89086

- VA Tennessee Valley Healthcare System (Adriana Hung, M.D., M.P.H.)

  1310 24th Avenue, South Nashville, TN 37212

- Washington DC VA Medical Center (Jack Lichy, M.D., Ph.D.)

  50 Irving St, Washington, D. C. 20422

- W.G. (Bill) Hefner VA Medical Center (Robin Hurley, M.D.)

  1601 Brenner Ave, Salisbury, NC 28144

- White River Junction VA Medical Center (Brooks Robey, M.D.)

  163 Veterans Drive, White River Junction, VT 05009

- William S. Middleton Memorial Veterans Hospital (Prakash Balasubramanian, M.D.)

  2500 Overlook Terrace, Madison, WI 53705

## Author Contributions

**Conceptualization:** Kyriacos Markianos, Saiju Pyarajan, Ronald Przygodzki.

**Data curation:** Kyriacos Markianos, Frederic Dong, Yunling Shi, Daniel Dochtermann, Uma Saxena, Poornima Devineni.

**Formal analysis:** Kyriacos Markianos, Frederic Dong, Bryan Gorman.

**Funding acquisition:** Sumitra Muralidhar, Rachel Ramoni, Philip Tsao, Saiju Pyarajan, Ronald Przygodzki.

**Investigation:** Kyriacos Markianos, Frederic Dong, Jennifer Moser, Sumitra Muralidhar, Rachel Ramoni, Philip Tsao, Saiju Pyarajan, Ronald Przygodzki.

**Methodology:** Kyriacos Markianos, Frederic Dong, Bryan Gorman, Saiju Pyarajan.

**Project administration:** Jennifer Moser, Sumitra Muralidhar, Rachel Ramoni, Philip Tsao, Saiju Pyarajan, Ronald Przygodzki.

**Resources:** Saiju Pyarajan.

**Software:** Kyriacos Markianos, Frederic Dong, Bryan Gorman, Yunling Shi, Uma Saxena.

**Supervision:** Kyriacos Markianos, Jennifer Moser, Sumitra Muralidhar, Rachel Ramoni, Philip Tsao, Saiju Pyarajan, Ronald Przygodzki.

**Validation:** Kyriacos Markianos, Frederic Dong, Bryan Gorman, Yunling Shi.

**Visualization:** Kyriacos Markianos, Frederic Dong.

**Writing – original draft:** Kyriacos Markianos, Frederic Dong, Ronald Przygodzki.

**Writing – review & editing:** Kyriacos Markianos, Frederic Dong, Bryan Gorman, Saiju Pyarajan, Ronald Przygodzki.

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
