## [Decision Letter · Decision Letter 0]

31 Oct 2022

PONE-D-22-23467Pharmacogenetic allele variant frequencies: An analysis of the VA’s Million Veteran Program (MVP) as a representation of the diversity in US population.PLOS ONE

Dear Dr. Markianos,

Thank you for submitting your manuscript to PLOS ONE. After careful consideration, we feel that it has merit but does not fully meet PLOS ONE’s publication criteria as it currently stands. Therefore, we invite you to submit a revised version of the manuscript that addresses the points raised during the review process.

 The reviewers are in favour to this work. However, several comments are raised and should be taken into consideration before it can be accepted for publication. Specifically, the underlying message of the LAI analysis was not clearly presented.

We look forward to receiving your revised manuscript.

Kind regards,

Hoh Boon-Peng, PhD

Academic Editor

PLOS ONE

Journal Requirements:

“This research is based on data from the Million Veteran Program, Office of Research and Development, Veterans Health Administration and was supported by award no. MVP000.”

“This work was supported by grant #MVP000 to SP from the Million Veteran Program (MVP) from the Veterans Affairs (VA) Office of Research and Development (ORD) (www.research.va.gov). The funders had no role in study design, data collection and analysis, decision to publish, or preparation of the manuscript.”

5. One of the noted authors is a group or consortium Million Veteran Program. In addition to naming the author group, please list the individual authors and affiliations within this group in the acknowledgments section of your manuscript. Please also indicate clearly a lead author for this group along with a contact email address.

Reviewers' comments:

Reviewer's Responses to Questions

**Comments to the Author**

1. Is the manuscript technically sound, and do the data support the conclusions?

Reviewer #1: Yes

Reviewer #2: Yes

2. Has the statistical analysis been performed appropriately and rigorously? 

Reviewer #1: No

Reviewer #2: Yes

3. Have the authors made all data underlying the findings in their manuscript fully available?

Reviewer #1: No

Reviewer #2: Yes

4. Is the manuscript presented in an intelligible fashion and written in standard English?

Reviewer #1: Yes

Reviewer #2: Yes

5. Review Comments to the Author

Reviewer #1: The authors analyzed the frequencies of pharmacogenetics relevant variants in a large US population. Regarding the frequencies of these variants, it is not clear whether those not mentioned in the manuscript are publicly available?

Other issues:

What is the message that the authors try to convey by conducting the LAI analysis?

The presentation of Fig.1 is neither clear nor informative, the authors should try to present in another way or just highlight some of the genes.

Why the column ‘LAI’ of ‘EUR’ in the table of Fig.2 is identical to ‘MVP HARE’, and why some of the SNVs do not have a value in LAI? I suggest the authors to switch the X and Y axis of the Fig.2.

In line 156, the authors referred Fig.2., does it mean that the SNVs in Fig.2 belong to the ones that in the CYP2D6*5 CNV region?

In line 182, it should be ‘fewer 2-copy samples’ rather than ‘fewer diploid samples’.

In Supplementary Fig.1, it is better to show the correlation coefficient and the P-value.

Reviewer #2: Overall, this is a well written manuscript and will add important information about pharmacogene allele frequencies in US populations.

Mostly, I have minor comments.

Genes should be in italics in accordance with HGVS nomenclature.

Please make sure that all abbreviations/acronyms are written out for clarity of the reader. Many are documented below.

line 32 -I prefer the term variants over polymorphisms unless all variants that are being discussed are greater than 1% frequency.

Line 88- first use of HARE, please write out acronym.

line 139 - while the use of the rsID is acceptable, for PGx readers, please also include *allele.

line 143 - please confirm if the drug name need to be capitalized.

line 144 - while the use of the rsID is acceptable, for PGx readers, please also include *allele.

line 153 - b-blockers, either write out beta or use Greek symbol as appropriate for publisher.

line 160 - first use of PCA, please write out acronym.

lines 181-182 - remove word clearly (that is up to the reader to determine) and "a lot"; make sure sentence reads well after revision.

lines 193, 393 - prefer the term variants to SNPs

Line 194 - first use of HIBAG, please write out acronym.

Line 199 - first use of SCAR, please write out acronym.

Line 199 - first use of SJS/TEN, please write out acronyms.

lines 206-207 - Please delete "something we hope to address in the future". If needed this would be for the discussion as a future research.

line 370 - QC'd is lab lingo, please write out

Line 374 - SHAPEIT appears to be some program, is this an acronym? Is there a source that needs to be cited?

lines 375, 397 - Minimac4 appears to be some program, is this an acronym? Is there a source that needs to be cited?

Lines 416-420 - check font. I am not sure that matters for formatting for publication.

6. PLOS authors have the option to publish the peer review history of their article (what does this mean?). If published, this will include your full peer review and any attached files.

Reviewer #1: No

Reviewer #2: No

---

## [Author Response · Author response to Decision Letter 0]

6 Jan 2023

All comments are included in the (1) Cover letter (2) Response to reviewers file.

---

## [Decision Letter · Decision Letter 1]

6 Feb 2023

Pharmacogenetic allele variant frequencies: An analysis of the VA’s Million Veteran Program (MVP) as a representation of the diversity in US population.

PONE-D-22-23467R1

Dear Dr. Markianos,

We’re pleased to inform you that your manuscript has been judged scientifically suitable for publication and will be formally accepted for publication once it meets all outstanding technical requirements.

Kind regards,

Hoh Boon-Peng, PhD

Academic Editor

PLOS ONE

Additional Editor Comments (optional):

Reviewers' comments:

Reviewer's Responses to Questions

**Comments to the Author**

1. If the authors have adequately addressed your comments raised in a previous round of review and you feel that this manuscript is now acceptable for publication, you may indicate that here to bypass the “Comments to the Author” section, enter your conflict of interest statement in the “Confidential to Editor” section, and submit your "Accept" recommendation.

Reviewer #1: All comments have been addressed

2. Is the manuscript technically sound, and do the data support the conclusions?

Reviewer #1: Yes

3. Has the statistical analysis been performed appropriately and rigorously? 

Reviewer #1: Yes

4. Have the authors made all data underlying the findings in their manuscript fully available?

Reviewer #1: Yes

5. Is the manuscript presented in an intelligible fashion and written in standard English?

Reviewer #1: Yes

6. Review Comments to the Author

Reviewer #1: The authors have provided explanations to my previous questions, I don't have any further questions at this round.

7. PLOS authors have the option to publish the peer review history of their article (what does this mean?). If published, this will include your full peer review and any attached files.

Reviewer #1: No

---

## [Editor Report · Acceptance letter]

15 Feb 2023

PONE-D-22-23467R1 

Pharmacogenetic allele variant frequencies: An analysis of the VA’s Million Veteran Program (MVP) as a representation of the diversity in US population. 

Dear Dr. Markianos:

I'm pleased to inform you that your manuscript has been deemed suitable for publication in PLOS ONE. Congratulations! Your manuscript is now with our production department. 

Kind regards, 

on behalf of

Professor Dr Hoh Boon-Peng 

Academic Editor

PLOS ONE